# ICLabAgent: Multi-Agent LMM Framework for Integrated Circuit Footprint Geometry Labeling

## Abstract

Integrated circuit (IC) footprint geometry labeling refers to the process of converting pin diagrams in IC datasheets into machine-readable geometric parameters. This task is critical in printed circuit board (PCB) design and component assembly, as accurate labeling ensures proper IC placement and reliable connectivity. The process is challenged by unstructured annotations, complex footprint arrangements, and abstract geometric diagrams, making fully automated labeling methods inadequate. Traditional EDA tools require heavy manual input and are slow. Existing automation methods, such as OCR or object detection, can extract text or simple shapes but fail to capture the implicit geometric relationships in IC diagrams, leaving the labeling task incomplete. Recent work has shown that end-to-end large multimodal models (LMMs) can perform IC geometry labeling. However, by treating the task as a black box, these methods are prone to shortcut learning and lack interpretability. In this work, we introduce ICLabAgent, the first multi-agent framework for fully automated IC footprint geometry labeling that explicitly models the workflow of expert engineers to produce more interpretable and reliable labeling outcomes. Furthermore, we present ICAgent-Instruct, the first dynamic planning and reasoning dataset tailored for IC footprint geometry labeling. Extensive experiments show that ICLabAgent improves overall accuracy by $10.3\%$ compared to the previous SOTA method and by $79.5\%$ compared to manual annotation. Despite using only simple supervised fine-tuning on a 7B model (Qwen2-VL-7B), ICLabAgent surpasses general-purpose LMMs such as GPT-5 (by $94.6\%$) and Gemini-2.5 Flash (by $378.8\%$).

## 1 Introduction

Integrated circuits (ICs) are miniaturized electronic systems with many interconnected components. IC pins are represented in footprint diagrams, which are often referred to as "Suggest Pads" or "Land Patterns" in IC datasheets. In printed circuit board (PCB) design, engineers must precisely label IC pin footprints, since this ensures correct component placement and reliable electrical connections. Many component vendors, such as Texas Instruments (Texas Instruments, 2025) and CUI (bel, 2025), provide footprint description diagrams within their IC datasheets. In addition, third-party libraries like Digi-Key (Electronics, 2025) and SnapEDA (SnapMagic Search, 2025) offer supplementary footprint resources. Despite the abundance of available IC datasheets and footprint diagrams (*e.g.*, over 13 million electronic products in Digi-Key), there remains a lack of efficient tools capable of converting these abstract geometric diagrams into machine-readable numerical descriptions. This gap slows down PCB design and manufacturing, as engineers must manually interpret diagrams, a process that is time-consuming, error-prone, and detrimental to design cycle speed and reliability (ASSEMBLY, 2021).

Traditional electronic design automation (EDA) tools require expert operation and lack automation for interpreting datasheet diagrams. As a result, they are inefficient and difficult to use for large-scale or rapid PCB design. Although deep-learning methods such as OCR (Du et al., 2024; Duan et al., 2025) and object detection (Jocher, 2020; Tan et al., 2020) are widely used for PDF text extraction and content understanding, they primarily focus on text comprehension and image description tasks. Consequently, they are inadequate for handling the complex text-symbol geometric understanding

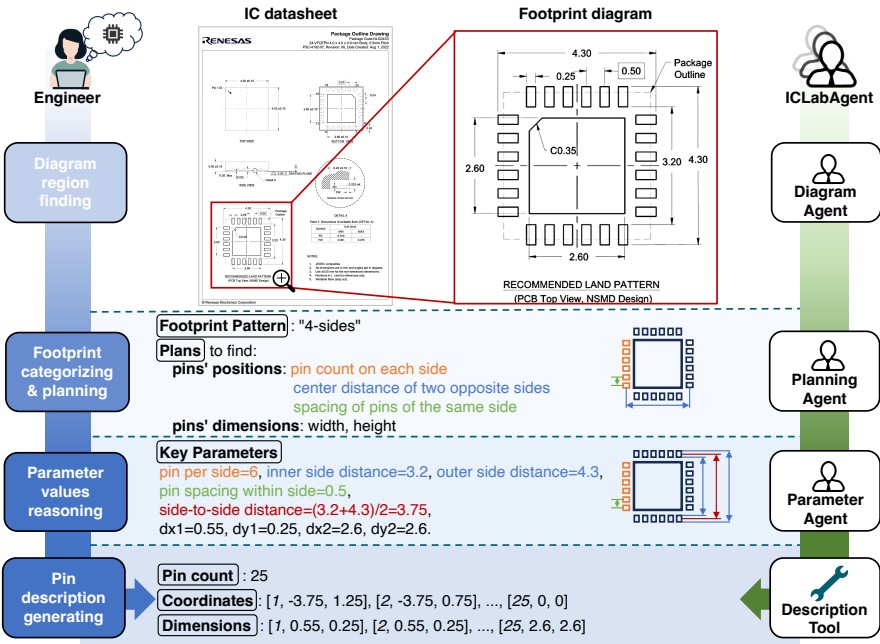

Figure 1: ICLABAGENT simulates the human expert engineer's reasoning workflow. Typical workflow for a PCB engineer to label IC footprint pin geometry information involves four sequential steps: **(1) searching for the diagram** in the datasheet, **(2) identifying the footprint type** (e.g., whether the IC has pins on 2 sides, 4 sides, or arranged in a grid) and **planning for the key parameters** to be extracted, **(3) searching for and calculating the key parameter values,** and **(4) manually entering the extracted key parameters into EDA tools** to generate footprint drawings with geometric information. ICLABAGENT directly mimics the above workflow and replaces each key step with a task-specific LMM agent, aiming at fully automated IC footprint labeling.

required for footprint diagrams, especially when implicit information is omitted. Large multimodal models (LMMs) advance this capability by deep reasoning over the text-image relationships and have demonstrated remarkable capabilities across a range of domains, including general-purpose reasoning (Hurst et al., 2024; Guo et al., 2025; Wang et al., 2024), geometric and mathematical reasoning (Lu et al., 2023; Masry et al., 2023; Gao et al., 2023), image understanding (Guo et al., 2024; Tzelepi & Mezaris, 2024), and medical diagnosis (Ikezogwo et al., 2023). Building on this progress, LLM4-IC8K (Wang et al., 2025a) introduces a two-stage training framework that fine-tunes an LMM for automated IC footprint geometry understanding and labeling. However, by treating the task as a black box, these end-to-end methods neglect the intermediate reasoning steps that PCB engineers naturally follow, such as locating the diagram, identifying IC types, and extracting parameters. These steps are essential for ensuring interpretability and reliability. In contrast, our multi-agent design explicitly mirrors this human workflow through diagram detection, type classification, parameter extraction, and description generation.

Recent advancements in agent-based systems (Park et al., 2023; Hong et al., 2023; Chen et al., 2024; Wu et al., 2024b) have demonstrated impressive capabilities across a wide range of domains, including strategic gameplay (Wang et al., 2023), complex task automation (Schick et al., 2023), multimodal content generation (Yang et al., 2023), medical diagnosis (Sun et al., 2025; Wang et al., 2025b), and embodied agent interaction (Schumann et al., 2024). Agent-based LMM systems enable dynamic task decomposition and strategic planning, making them particularly well-suited for domains that involve expert knowledge, complex reasoning workflows, and multi-step decision-making pipelines. However, no prior work has applied such frameworks to IC geometry labeling, leaving a gap that our work addresses.

Motivated by the natural reasoning process employed by PCB engineers during IC footprint labeling tasks, we propose **ICLABAGENT, a novel framework that enables step-by-step dynamic reasoning, explicitly simulating the expert workflow used by real-world PCB engineers**. ICLABAGENT employs multiple task-specific agents to construct a multi-stage workflow for generating pin description labels of IC footprints, as illustrated in Figure 1. Our main contribution of this article in advancing agent-based IC footprint labeling are fourfold:

1. **We introduce ICLABAGENT, the first multi-agent framework that emulates the IC footprint understanding and labeling workflow of expert PCB engineers** by incorporating IC footprint classification, dynamic planning, and key parameter extraction through multi-step reasoning.

2. **We develop ICAGENT-INSTRUCT, the first dynamic planning and reasoning dataset specifically designed for IC footprint geometry labeling**. This dataset is tailored to fine-tune LMMs with planning strategies and geometric understanding reflective of expert-level PCB engineering knowledge. All datasets will be publicly available upon acceptance.

3. **We propose a multi-stage training strategy that sequentially fine-tunes three task-specific LMM agents**, each specializing in a distinct subtask: footprint diagram localization, footprint classification and parameter planning, and key parameter extraction.

4. **We evaluate our framework extensively on the IC footprint geometry labeling benchmark and each workflow stage**. The results demonstrate our approach's effectiveness in providing interpretable and practical reasoning for IC footprint labeling. Our method uses only SFT training with the lightweight base model Qwen2-VL-7B (Wang et al., 2024). In contrast, state-of-the-art general LMMs, such as GPT-5 (OpenAI, 2025) and Gemini 2.5 Flash (Comanici et al., 2025), fail to complete the labeling task.

## 2 RELATED WORK

### 2.1 LMM AGENTS

LMM agents are intelligent systems powered by LMMs, capable of perceiving the environment, reasoning about goals, and executing actions. These capabilities enable them to autonomously perform complex tasks traditionally requiring human expertise, while adapting to task-specific requirements (Luo et al., 2025). Recent advances in LMM agents have demonstrated notable success in simulating domain-expert workflows across a range of applications, including strategic game-play (Wang et al., 2023), software development (Yang et al., 2023; Qian et al., 2023), creative content generation (Yang et al., 2022), embodied interaction (Schumann et al., 2024), and clinical diagnosis (Sun et al., 2025; Wang et al., 2025b). Central to their effectiveness is the perceiving–planning–reasoning–execution architecture, which equips these agents to handle dynamic environments and resolve multi-step, case-specific problems. LMM agents have shown growing potential in the PCB engineering industry. Most existing applications focus on EDA, encompassing domain-specific assistant chatbots (Liu et al., 2023; Han et al., 2023; Chen et al., 2025), hardware description language (HDL) generation (Wu et al., 2024a; Chang et al., 2023), and code verification and analysis (Orenes-Vera et al., 2021). However, these efforts primarily address the macroscopic aspects of PCB design, often neglecting the foundational components, such as detailed IC footprint information. These footprints are typically embedded within lengthy datasheet PDFs in the form of abstract geometric diagrams, which still require significant manual effort to convert into structured numeric descriptions. We introduce the first LMM agent system devoted to addressing the IC footprint labeling problem, laying the foundation for automating various downstream EDA tasks.

### 2.2 AUTOMATED IC FOOTPRINT LABELING

Traditional PCB component geometry generation is a labor-intensive and time-consuming process that involves manually interpreting datasheets, creating footprints, generating symbols, and mapping signals (Ni et al., 2020; hun, 2021). As modern designs often involve hundreds of components, this manual approach becomes a significant bottleneck (Martens, 2022), susceptible to inconsistencies, human error, and outdated libraries due to frequent specification updates (Teel, 2023; Abraham, 2025). These challenges highlight the critical need for automated, data-driven solutions to enhance both the efficiency and reliability of PCB design workflows. Existing PCB labeling methods primarily focus on the segmentation or classification of IC footprints (Ni et al., 2020; Yang et al., 2024). However, these approaches fail to address the geometric information of individual IC pins, limiting the automated labeling process to a higher level, overlooking the intricate details of pin-level geometry. Although object detection methods (Jocher, 2020; Tan et al., 2020) can assist in counting IC pins and estimating their relative sizes, they struggle with footprint diagrams that omit implicit information. Optical Character Recognition (OCR) methods (Du et al., 2024; Duan et al., 2025) can

extract textual data from datasheet images, but cannot interpret the physical or geometric meaning of numerical labels. Some works combine object detection and OCR for diagram object detection (Hu et al., 2023) or for converting legacy schematic diagrams (Nurminen et al., 2020). However, these methods do not address the gap between the extracted annotations and the underlying geometric knowledge. LLM4-IC8K (Wang et al., 2025a) introduces a two-stage training framework that fine-tunes an LMM for automated IC footprint geometry understanding and labeling. However, this end-to-end approach treats the IC footprint labeling task as a black-box problem, neglecting the intermediate reasoning processes employed by expert PCB engineers. In contrast, our framework introduces step-by-step dynamic reasoning, explicitly simulating the workflow of real-world PCB engineers, which leads to more interpretable and reliable footprint labeling performance.

## 3 IC GEOMETRY UNDERSTANDING PROBLEM

### 3.1 IC FOOTPRINT LABELING WORKFLOW

Understanding the IC diagrams is fundamental to effective PCB design and accurate IC module mounting. The geometry of IC footprints encompasses the spatial configuration of IC pins and the dimensional specifications of each pin as represented in technical schematics. Within the PCB design and manufacturing industry, the arrangement and dimensions of an IC module's pin footprints critically influence both its physical placement on the board and the integrity of its electrical connections, thereby ensuring the proper functionality of the circuit.

The problem of understanding IC geometry labeling can be delineated into three primary components: (1) determining the total number of pins in the IC diagram, (2) analyzing the spatial arrangement (i.e., coordinates) of the pins, and (3) discerning the shape and dimensions of each individual pin. While experienced engineers may be able to estimate the number of pins in low-pin-count IC footprint diagrams through intuition, the precise coordinates and dimensions of the pins cannot be ascertained instantaneously. Furthermore, IC modules often comprise over a hundred pins, rendering manual and immediate pin counting impractical. As a result, engineers typically adopt a step-by-step approach to deconstruct the diagrams and extract key parameters that facilitate accurate identification of IC footprints.

When identifying an IC footprint, an engineer typically begins by determining the number of pins depicted in the diagram. This is achieved by analyzing the geometric structure of the diagram, such as counting the number of rows and columns in a uniform pin array or the number of pins per edge in a centrosymmetric ring configuration (as illustrated in Figure 1 ). Subsequently, the engineer interprets the spatial relationships among the pins based on the geometric annotations provided in the diagram, including row spacing, column spacing, and distances from the module center to the edges. The engineer then assesses the dimensions of individual pins using the labeled measurements, such as height and width for rectangular pins or diameter for circular pins. Finally, by summarizing this geometric information, the engineer determines the pin count, spatial coordinates, and dimensions of the IC footprint. In this work, we propose a multi-agent framework designed to emulate the step-by-step reasoning process of an expert engineer, thereby enhancing the geometric understanding of LMMs when interpreting IC footprint diagrams.

### 3.2 CHALLENGES

A primary challenge for LMMs in interpreting IC footprint diagrams lies in accurately identifying the key parameters presented in the diagrams and comprehending the relationships between numeric labels and graphical symbols. Often, these critical parameters are not explicitly provided and require additional reasoning to infer. For instance, as illustrated in Figure 2, the center-to-center distance between two pin rows is not directly labeled. Instead, it must be derived from the relationships among the outer-edge distance, the inner-edge distance of the pins, and their spatial arrangement. The agent must also be capable of adapting its reasoning workflow to accommodate different types of IC modules. Due to the variability in the spatial arrangement of pins across module types, the key parameters that need to be extracted, as well as the tools required for generating accurate descriptions of these parameters, differ accordingly. Therefore, it is essential to design reasoning workflows that account for the diverse types of IC pin arrangements and support multi-step parameter inference to enable human-like reasoning capabilities in LMMs for IC footprint geometry understanding.

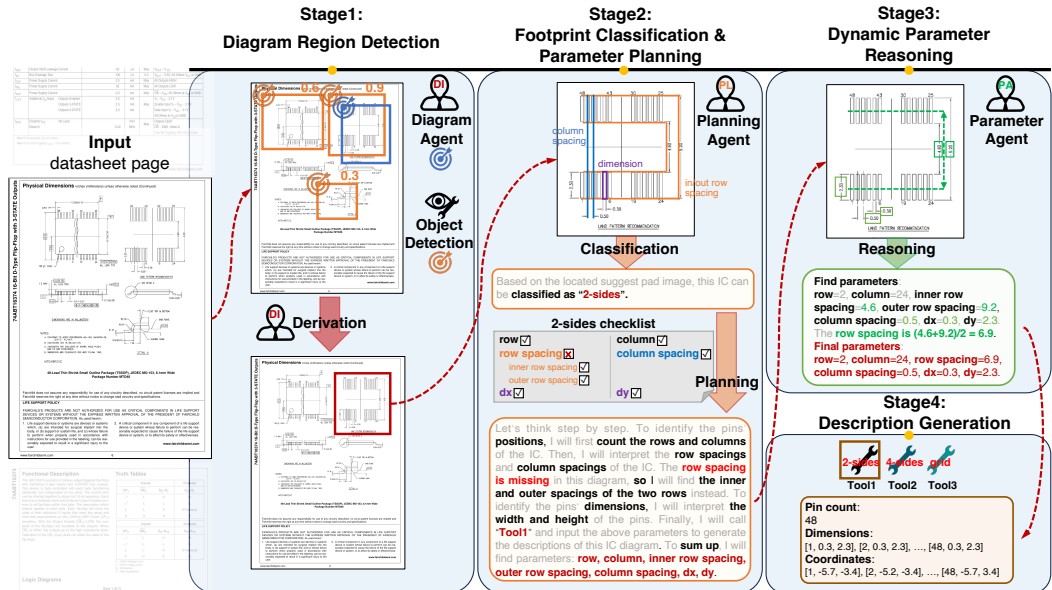

Figure 2: Overview of the proposed agent framework ICLABAGENT. The reasoning pipeline includes four sequential stages. First, the **Diagram Agent** detects and locates the region of the footprint diagram in the datasheet page, and sends the region information to the Planning Agent. The **Planning Agent** then analyzes the arrangement pattern of the IC and generates instructions to guide the key parameter extraction process. After receiving instructions from the Planning Agent, the **Parameter Agent** finds accessible key parameters in the diagram and calculates implicit key parameters based on given instructions. Finally, ICLABAGENT calls pin **description generation** tools based on the IC type classification to generate a detailed geometric description for each IC pin.

# 4 METHODS

## 4.1 REASONING WORKFLOW

As described in Section 3.1, human engineers typically follow a four-step workflow to label IC footprints. To replicate this process, we design ICLABAGENT as a multi-agent system. Each agent corresponds to one of these steps, ensuring that the overall workflow mirrors the reasoning process of real engineers while remaining interpretable: (1) identifying the region containing the target IC footprint diagram, (2) recognizing the footprint's arrangement type and determining the key parameters required to infer the coordinates and dimensions of the pins, (3) locating and computing these key parameters based on the numeric annotations provided in the diagram, and (4) generating a comprehensive description of the pins using the extracted parameters. To emulate expert-level understanding of IC footprint geometry labeling, we develop a multi-agent system capable of dynamic diagram region detection, classification of IC pin arrangement types, strategic planning for key parameter extraction, and geometric reasoning over the identified parameters. As illustrated in Figure 2, the IC footprint geometry labeling workflow can be arranged into four key stages with three task-specific agents:

**1) Diagram Region Detection:** When given a PDF page of an IC datasheet containing the IC footprint diagram, ICLABAGENT adopts the **Diagram Agent** to identify the region of the target diagram described in a four-element number pair $(x, y, dx, dy)$, where $x$, $y$ denote the coordinates of the upper left corner of the diagram region, and $dx$, $dy$ denote the width and height of the region in proportions of the datasheet image sizes. The Diagram Agent first invokes an object-detection tool to identify candidate regions of interest (ROIs) corresponding to the footprint diagrams, each with an associated confidence score (orange boxes in Stage 1 in Figure 2). It then cross-validates these candidates using its own diagram-detection judgments (blue box) and fuses the results to yield the final detection region (red box). The detected diagram region enables downstream agents to focus on the relevant diagram, minimizing interference from surrounding, unrelated diagrams and text.

**2) Footprint Classification and Parameter Planning:** After identifying the diagram region, ICLABAGENT deploys the **Planning Agent**, which processes region-specific prompts alongside

Figure 3: We categorize IC packages into four groups based on pin distribution: **dual-side packages** (e.g., SOP, SOIC), **quad-side packages** (e.g., QFP, QFN), **grid array packages** (e.g., BGA, LGA), and **others** (e.g., SOT, TO). For simplicity, we refer to them as "2-sides", "4-sides", "grid", and "other" throughout this paper. "2-sides", "4-sides", and "grid" footprints are symmetric and can be structurally described by several key parameters, while "other" footprints are asymmetric but often contain fewer pins, which require a distinct description for each pin.

the diagram image. The Planning Agent classifies the IC footprint type based on the spatial arrangement of its pins. To ensure representativeness while maintaining generality, we categorize common IC footprints into four classes based on their pin arrangement patterns (Ayodele, 2024), as illustrated in Figure 3: 2-sides, 4-sides, grid, and other. The first three categories account for over $80\%$ of all IC footprints (as detailed in Appendix A.2) and collectively represent the majority of standard IC footprint templates. Next, the Planning Agent identifies key parameters necessary to infer pin coordinates and dimensions based on the previously determined arrangement type. It follows an IC-type-specific checklist derived from the classification result, which includes all key parameters that may appear in the IC diagram of the specified type. The agent evaluates both existing parameters and those missing but essential for accurate footprint identification. Specifically, the Planning Agent first identifies the key parameters explicitly annotated in the diagram. For parameters not directly provided, the agent performs parameter reasoning to determine how these values can be derived from the available annotations. Finally, the agent compiles a summary of all relevant key parameters, which guides the downstream parameter extraction process. IC footprints categorized as "other" typically exhibit irregular pin arrangements and usually contain fewer than 20 pins. As a result, identifying pin coordinates and dimensions is less complex. Therefore, we adopt a direct approach to describe the pin count, coordinates, and dimensions for footprints in this category.

**3) Dynamic Parameter Reasoning:** Under the guidance provided by the Planning Agent, the **Parameter Agent** identifies and interprets the key parameters expressed numerically within the diagram. For parameters that are not explicitly annotated, the Parameter Agent applies the reasoning instructions specified by the Planning Agent to compute the required values. Finally, it compiles the extracted and inferred parameters into a type-specific standardized format.

**4) Description Generation:** Based on the identified pin arrangement type, ICLABAGENT invokes the appropriate description generation tool corresponding to "2-sides", "4-sides", or "grid" footprints. These tools take the standardized key parameters as input and produce pin descriptions aligned with the three IC footprint labeling tasks outlined in Section 3.1: determining the pin count, computing the pin coordinates, and specifying the pin dimensions. For "other" footprints, the pin descriptions are directly inferred through the collaborative processing of the Planning Agent and the Parameter Agent. Thus, no description generation tool is required for this category.

## 4.2 DATASET CONSTRUCTION

To effectively train our agent-based model to simulate the complex reasoning workflow involved in IC footprint understanding, To train ICLABAGENT, we construct the ICAGENT-INSTRUCT dataset. It contains real datasheet diagrams with expert annotations and contains three subsets aligned with our agents' subtasks: (1) diagram detection, (2) parameter planning, and (3) parameter extraction. Each subset reflects the reasoning steps that engineers use, ensuring that **training data encodes expert-level strategies rather than only end results**.

We build upon the real-world datasheets provided in the ICGEO8K dataset proposed by Wang et al. (2025a), which includes 4,138 IC footprint entries. To ensure data quality, we filter out diagrams

that are mislabeled, occluded, or incorrectly classified as 2D footprint diagrams. After this cleaning process, we obtain a total of 3,737 valid sequences, comprising 3,337 training samples from the original real-world training subset and all 400 test samples (excluded during the fine-tuning process) from ICGEO8K. The reasoning datasets are constructed using the datasheet page images and their corresponding pin description labels. The data generation process strictly adheres to the reasoning workflow outlined in Section 4.1, and incorporates additional expert annotations to improve data quality. In the end, we construct three novel datasets based on the real-world samples: the **Diagram Detection Dataset**, the **Parameter Planning Dataset**, and the **Parameter Extraction Dataset**. The three subsets are detailed in Appendix A.1.

### 4.3 MODEL TRAINING

To implement an effective multi-agent framework, we adopt a three-step training strategy aimed at developing specialized functional agents. At the core of each agent lies a powerful vision-language model, Qwen2.5-VL. We fine-tune this multimodal model on the three task-specific reasoning datasets described in the previous section, resulting in three dedicated models tailored to support the corresponding workflow agents.

We begin by training the Diagram Agent using the Diagram Detection Dataset to enable accurate localization of IC footprint regions. We also train an object detection tool using the same Diagram Detection Dataset to acquire IC ROI candidates. Next, the Planning Agent is trained on the Parameter Planning Dataset to develop its ability to classify footprint types and plan procedures to identify key parameters. Finally, the Parameter Agent is trained using the Parameter Extraction Dataset, allowing it to extract and compute standard key parameters based on the Planning Agent outputs. At each training stage, we incorporate chain-of-thought (CoT) training to promote step-by-step reasoning and ensure the models develop a robust and interpretable understanding of the tasks.

## 5 EXPERIMENTS

### 5.1 EXPERIMENT SETUP

**Implementation Details.** As described in Section 4.3, we adopt Qwen2-VL (Wang et al., 2024) as the visual question answering backbone for our agents. Specifically, we utilize the 7B version (Qwen2-VL-7B) as the base model for all task-specific agents in this work. All experiments are conducted using 2 NVIDIA A100-40GB GPUs. For detailed training configurations and hyperparameters, please refer to Appendix A.5.

**Evaluation Metrics.** We adopt the same evaluation metrics as LLM4-IC8K: $IoU_{IC}$ measures the Intersection over Union (IoU) ratio between predicted and ground-truth footprints; Mean Absolute Error ($MAE$) and Root Mean Square Error ($RMSE$) measure pin count accuracy (*task 1*); $d_{pin}$ measures coordinate precision (*task 2*); $IoU_{pin}$ measures pin dimension accuracy (*task 3*).

For intermediate agents, we further report the following metrics. For **Stage 1 (diagram detection)**, $IoU_{diagram}$ measures the IoU ratio between predicted and ground-truth diagram bounding boxes. For **Stage 2 (classification and planning)**, Type Classification Accuracy ($TCA$) measures the footprint type classification accuracy, and Key Parameter Accuracy ($KPA$) measures whether the Planning Agent identifies the correct set of key parameters. For **Stage 3 (parameter reasoning)**, Parameter Completion rate ($PC$) measures the recovery rate of required parameters, while $MAE$ and $RMSE$ capture the numerical accuracy of predicted values.

**Evaluation Baseline.** We evaluate our framework on ICGEOQA, a benchmark consisting of 400 representative real-world IC datasheet pages, and compare its performance against the current state-of-the-art method for IC footprint labeling, LLM4-IC8K (Wang et al., 2025a). For a fair comparison, all baselines were trained and tested under the same hardware settings. **General-purpose LMMs** (e.g., GPT-4o, GPT-5, Gemini 2.5 Flash) were evaluated in zero-shot settings, while the **specialized model** (*i.e.*, LLM4-IC8K) used its reported fine-tuning pipeline. A **manual industry baseline** is also computed from real-world engineer annotation data (detailed in Appendix A.3) to serve as the accuracy standard for industrial human experts.

Table 1: Comparison of QA performance with general LLMs on 3 tasks in mean + std format. The best-performing method is **in-bold**, and the second-best is underlined.

| Methods | Overall ($IoU_{IC}$ %)↑ | Task 1 | | Task 2 ($d_{pin}$)↓ | Task 3 ($IoU_{pin}$ %)↑ |
|---|---|---|---|---|---|
| | | MAE↓ | RMSE↓ | | |
| **General Large Multimodal Models** | | | | | |
| GPT-4o Hurst et al. (2024) | $11.1 \pm 0.4$ | $8.21 \pm 0.47$ | $23.04 \pm 0.22$ | $4.01 \pm 0.02$ | $45.6 \pm 0.3$ |
| GPT-5 OpenAI (2025) | $40.6 \pm 1.1$ | $0.39 \pm 0.04$ | $4.13 \pm 0.19$ | $4.18 \pm 0.22$ | $65.4 \pm 0.1$ |
| Gemini 2.0 Mallick & Kilpatrick (2025) | $4.5 \pm 0.1$ | $1.84 \pm 0.40$ | $7.87 \pm 0.85$ | $18.27 \pm 0.45$ | $57.3 \pm 0.4$ |
| Gemini 2.5 Flash (Comanici et al., 2025) | $16.5 \pm 0.3$ | $11.13 \pm 0.70$ | $35.91 \pm 0.21$ | $3.56 \pm 0.15$ | $35.8 \pm 0.6$ |
| Gemma3-12B Team et al. (2025) | $5.8 \pm 0.1$ | $8.84 \pm 0.42$ | $26.09 \pm 0.85$ | $5.13 \pm 0.13$ | $21.5 \pm 0.5$ |
| Qwen2.5-VL-7B (Bai et al., 2025) | $5.4 \pm 0.2$ | $24.07 \pm 0.13$ | $46.58 \pm 0.43$ | $4.38 \pm 0.17$ | $6.2 \pm 0.2$ |
| **Manual Industry Baseline** | | | | | |
| Manual EDA | 44 | - | - | 2.98 | 58 |
| **IC Footprint Labeling Large Multimodal Models** | | | | | |
| LLM4-IC8K (Wang et al., 2025a) | $71.6 \pm 0.5$ | **$0.35 \pm 0.07$** | **$2.81 \pm 0.08$** | $1.11 \pm 0.02$ | $88.0 \pm 0.3$ |
| **ICLABAGENT** | **$79.0 \pm 0.2$** | $0.37 \pm 0.05$ | $3.25 \pm 0.14$ | **$1.07 \pm 0.03$** | **$95.6 \pm 0.2$** |

Table 2: Comparing performances on three stages with general-purpose LMMs.

| Methods | Stage1 | Stage2 | | | Stage3 | |
|---|---|---|---|---|---|---|
| | $IoU_{diagram}$ %↑ | TCA %↑ | KPA %↑ | PC %↑ | MAE↓ | RMSE↓ |
| GPT-5 OpenAI (2025) | $2.9 \pm 0.6$ | $79.5 \pm 0.2$ | $93.1 \pm 0.5$ | $99.8 \pm 0.4$ | $1.94 \pm 0.06$ | $4.18 \pm 0.38$ |
| Gemini 2.5 flash Comanici et al. (2025) | $3.9 \pm 0.7$ | $84.6 \pm 0.2$ | $86.6 \pm 0.4$ | $80.4 \pm 0.1$ | $1.50 \pm 0.04$ | $3.55 \pm 0.17$ |
| Gemma3-12B Team et al. (2025) | $11.4 \pm 0.3$ | $33.9 \pm 0.5$ | $44.9 \pm 0.1$ | $61.1 \pm 0.8$ | $2.84 \pm 0.17$ | $5.04 \pm 0.51$ |
| Qwen2.5-VL-7B Bai et al. (2025) | $3.0 \pm 0.3$ | $28.3 \pm 0.3$ | $24.1 \pm 0.4$ | $81.6 \pm 0.5$ | $2.21 \pm 0.28$ | $10.13 \pm 0.15$ |
| **ICLABAGENT** | **$70.2 \pm 0.4$** | **100** | **$97 \pm 0.1$** | **$82.3 \pm 0.4$** | **$0.22 \pm 0.03$** | **$0.39 \pm 0.06$** |

## 5.2 OVERALL COMPARISON

We evaluate the overall performance of our framework and compare it against both the baseline method and several general-purpose LMMs. As shown in Table 1, ICLABAGENT achieves an overall $IoU_{IC}$ of 79.0% on the ICGEOQA benchmark, substantially outperforming both general-purpose LMMs and the previous state-of-the-art IC footprint labeling model. While ICLABAGENT shows sub-optimal performance in IC pin counting compared to LLM4-IC8K, the differences are negligible, with an average error of less than 1 pin, and it still outperforms all general LMMs. Additionally, ICLABAGENT demonstrates superior capability in pin coordinate and dimension estimation, enhancing the overall geometric understanding accuracy by 10.3% compared to the SOTA task-specific LMM and surpassing the industry manual baseline by 79.5%. This is due to the simplicity and stability of the framework, which focuses on handling concise key parameters rather than directly estimating the complex and lengthy pin geometry descriptions. ICLABAGENT encompasses expert-level knowledge for IC footprint identification and geometric reasoning, while emulating the step-by-step reasoning workflow of PCB engineers to support a dynamic labeling process. By adhering closely to the procedural logic employed by human experts, ICLABAGENT mitigates the risks of hallucinations and shortcut learning commonly observed in end-to-end fine-tuned LMMs. As a result, **ICLABAGENT achieves superior performance in IC footprint geometry understanding and labeling.**

## 5.3 COMPONENT EVALUATION

To demonstrate the reliability of the intermediate reasoning processes, we evaluate the performance of each task-specific agent within the decision-making pipeline, shown in Table 2.

**Evaluation on Diagram Region Detection.** We evaluate the performance of the Diagram Agent on the ICGEOQA benchmark and compare it with general-purpose LMMs using a one-shot learning setup. Our Diagram Agent achieves an $IoU_{diagram}$ of 70.2% in accurately locating diagram regions within datasheet pages, significantly outperforming the baseline models. General LMMs face challenges in accurately locating the correct diagram regions without specialized training or extensive prior knowledge. This result highlights the importance of training a dedicated agent for diagram region detection in IC footprint understanding.

**Evaluation on Footprint Classification and Parameter Planning.** Next, we evaluate the performance of the Planning Agent on footprint arrangement classification and parameter planning. The Planning Agent achieves a TCA of 100% and a KPA of 97%, demonstrating its effectiveness in accurately identifying footprint arrangement types and generating precise, footprint-specific plans for key parameter extraction. GPT-5 and Gemini 2.5 Flash also show fine performances in identifying footprint types and parameter recognition, expressing great potential in IC geometric understanding.

**Evaluation on Footprint Classification and Parameter Planning.** Next, we evaluate the performance of the Planning Agent on footprint arrangement classification and parameter planning. The Planning Agent achieves a TCA of $100\%$ and a KPA of $97\%$, demonstrating its effectiveness in accurately identifying footprint arrangement types and generating precise, footprint-specific plans for key parameter extraction. Despite their failure in diagram region detection, GPT-5 and Gemini 2.5 Flash still demonstrate strong performance in identifying footprint types and recognizing parameters. This suggests that general LMMs typically process information at the whole-image level, while ICLABAGENT leverages zoomed-in views for more precise detail extraction.

**Evaluation on Dynamic Parameter Reasoning.** The Parameter Agent achieves a PC of $82.3\%$ for key parameter recovery, with an MAE of $0.22$ and an RMSE of $0.39$ for parameter value recognition and reasoning. While the Parameter Agent does not recover all parameters specified in the instructions, its precise numeric value retrieval and parameter calculation process significantly reduces errors in the overall parameter extraction. These results show the importance of training a dedicated agent for accurate parameter extraction and multi-step geometric reasoning.

## 5.4 STAGE ABLATION

We conduct an ablation study to evaluate the contribution of each stage in our reasoning workflow, as presented in Table 3. In one setting, diagram region localization (Stage1) is removed, requiring downstream agents to infer the location of footprint diagrams without explicit guidance. In another setting, footprint classification and parameter planning (Stage2) are excluded, and the downstream agent is tasked with identifying and extracting key pa-

Table 3: Ablation test on reasoning stages.

| Methods | $IoU_{IC}\%\uparrow$ |
|---|---|
| **Full stages (ICLABAGENT)** | **79.0±0.2** |
| without stage1 | 72.3±0.3 |
| without stage2 | 70.2±0.2 |
| without stage1&stage2 | 68.2±0.4 |
| end-to-end | 65.7±0.1 |

rameters independently. The third setting excludes both Stage1 and Stage2 and directly infers key parameter values from datasheet pages without structured guidance. Additionally, we compare with a fully end-to-end baseline, following the training strategy of LLM4-IC8K, which directly generates pin descriptions without explicitly reasoning through intermediate key parameters.

Removing **Stage 1 (diagram detection)** reduces $IoU$ by $8.5\%$, showing that precise region localization is crucial. Removing **Stage 2 (classification and planning)** drops performance by $11.1\%$, confirming that explicit planning avoids confusion during extraction. With only **Stage 3 (parameter reasoning)**, irrelevant information and $IoU_{IC}$ falls by $13.7\%$. Interestingly, we can see that the performance of the **Stage-3-only** still exceeds that of the black-box **end-to-end LMM** approach. This is attributed to the simplicity and structure of the parameter extraction process, which enables the LMM to recover key geometric parameters more effectively than directly estimating the coordinates and dimensions of each pin. These results demonstrate that each stage contributes substantially, and that decomposed reasoning is more reliable than end-to-end black-box training.

## 6 CONCLUSION

We introduced ICLABAGENT, a multi-agent framework for fully automated IC footprint geometry labeling. ICLABAGENT simulates the labeling workflow of expert PCB engineers by integrating IC footprint classification, dynamic planning, and key parameter extraction through multi-step reasoning. The framework comprises three task-specific agents responsible for: (1) diagram region detection, (2) footprint classification and parameter planning, and (3) dynamic parameter reasoning. To empower the dynamic reasoning capabilities of ICLABAGENT, we developed a novel dataset, ICAGENT-INSTRUCT, which encapsulates expert-level decision-making processes and reasoning strategies tailored for IC footprint labeling. We further proposed a multi-stage training pipeline to fine-tune the three LMM agents on their respective tasks. Experimental results show that our multi-agent framework achieves a labeling IoU accuracy of $79.0\%$, surpassing the manual baseline by $79.5\%$, thereby demonstrating the effectiveness and reliability of our approach.

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

# A  APPENDIX

## A.1  DATASET CONSTRUCTION

To effectively train our agent-based model to simulate the complex reasoning workflow involved in IC footprint understanding, we construct the ICAGENT-INSTRUCT dataset, comprising three distinct subsets tailored to the training requirements of the three task-specific agents.

We build upon the real-world datasheets provided in the ICGEO8K dataset proposed by Wang et al. (2025a), which includes 4,138 IC footprint entries. To ensure data quality, we filter out diagrams that are mislabeled, occluded, or incorrectly classified as 2D footprint diagrams. After this cleaning process, we obtain a total of 3,737 valid sequences, comprising 3,337 training samples from the original real-world training subset and all 400 test samples (excluded during the fine-tuning process) from ICGEO8K. The reasoning datasets are constructed using the datasheet page images and their corresponding pin description labels. The data generation process strictly adheres to the reasoning workflow outlined in Section 4.1, and incorporates additional expert annotations to improve data quality. In the end, we construct three novel datasets based on the real-world samples: the **Diagram Detection Dataset**, the **Parameter Planning Dataset**, and the **Parameter Extraction Dataset**. The three subsets are detailed as follows:

**1) Diagram Detection Dataset:**  We begin by identifying the regions containing footprint diagrams within the datasheet pages. To facilitate this process, we use Label Studio (Tkachenko et al., 2020-2025) to annotate the bounding boxes corresponding to the footprint diagram regions. Each footprint diagram bounding box is manually annotated with (1) the coordinates of its upper-left corner and (2) its width and height. This annotated dataset is used to train the Diagram Agent to accurately identify the regions containing IC footprint diagrams.

**2) Parameter Planning Dataset:**  The core reasoning capability of ICLABAGENT lies in the workflow planning conducted by the Planning Agent. To train the LMM to perform strategic parameter understanding in a manner that mimics expert human reasoning, we propose a four-step dataset construction procedure. First, we manually annotate the arrangement types of the identified footprint diagrams. Next, based on the classified type, annotators are presented with a set of candidate key parameters and are instructed to fill in only those parameters that are explicitly provided within the footprint diagram. Then, we identify whether any standard key parameters are missing from the diagram (*e.g.*, a missing center distance) but can be computed using the available labeled parameters (*e.g.*, by averaging the outer-edge and inner-edge distances). This reasoning process is then added into the planning workflow. If the provided parameters are insufficient to derive the standard key parameters, the model is instructed to examine the surrounding context within the datasheet page

to infer the missing information. Finally, the Planning Agent is instructed to summarize all explicit parameters present in the diagram that are involved in the planning process, providing a structured input for the downstream Parameter Agent. The aforementioned planning steps, combined with the corresponding datasheet images and annotated diagram bounding boxes, constitute the Parameter Planning Dataset.

**3) Parameter Extraction Dataset:** The workflow of the Parameter Agent is guided by the output parameter sets generated by the Planning Agent. For each data sample, the Parameter Agent is tasked with extracting the standard key parameters corresponding to the identified footprint arrangement type. The outputs from the Planning Agent are first examined to identify any missing standard key parameters. If standard key parameters are missing, the Parameter Agent initiates a reasoning step to compute them using the available non-standard parameters. Finally, the Parameter Agent consolidates the results into a complete set of standard key parameters. For IC footprints classified as "other", we utilize the data labels from ICGEO8K to extract their pin count, pin coordinates, and pin dimensions. ICGEO8K data labels are also employed to identify missing pins in grid-arranged footprints with large pin counts. Additionally, the labels from ICGEO8K serve as a reference for cross-validating the accuracy of our annotations, thereby ensuring the reliability and consistency of the dataset for training effective models.

In summary, we construct three novel datasets derived from the real-world ICGEO8K dataset, each designed to capture different aspects of the dynamic reasoning process involved in IC footprint geometry labeling.

## A.2 DATA STATICS

Our dataset, ICAGENT-INSTRUCT, is constructed based on ICGEO8K (Wang et al., 2025a). Consequently, the data distribution of ICAGENT-INSTRUCT mirrors that of the real-world subset of ICGEO8K, with minor modifications and selections applied to satisfy our training requirements.

ICGEO8K includes 10 IC package types: SOP, SOIC, DFN, QFP, QFN, LGA, BGA, SOT, TO, and Special. These packages can be grouped into the four categories defined in this work based on their footprint arrangements: SOP and SOIC are classified as "2-sides" ICs, DFN, QFP, and QFN as "4-sides" ICs, LGA and BGA as "grid" ICs, and SOT, TO, and Special as "other" ICs.

The type distributions of the datasets are shown in Figure A.1. In addition to ICAGENT-INSTRUCT and ICGEOQA, we present the IC type distribution of the 200K dataset, which comprises a large number of IC part entries (over 200K) collected from Digi-Key (Electronics, 2025). The type distribution of ICAGENT-INSTRUCT aligns closely with that of this extensive IC collection, demonstrating its representativeness of the real-world distribution within the IC community.

## A.3 MANUAL INDUSTRY BASELINE

Since the EDA description files corresponding to the benchmark ICs in ICGEO8K are directly sourced from industrial PCB libraries, they are considered sufficient to meet industry standards. However, engineers do not always strictly follow the specifications depicted in footprint diagrams, leading to discrepancies between datasheet diagrams and actual EDA annotations. Therefore, the accuracy of aligning manual EDA descriptions with footprint diagram annotations serves as a practical baseline for evaluating industrial-standard functional labeling.

The overall low performance of manual labeling indicates that engineers exercise a certain degree of flexibility when specifying pin geometries—for example, adjusting pin coordinates as long as relative positions conform to the Suggest Pad diagram, or modifying pin dimensions within the suggested regions. However, these non-standardized labeling practices hinder the standardization of PCB design procedures, potentially resulting in IC part violations and operational malfunctions. In contrast, ICLABAGENT strictly adheres to the footprint geometries specified in the Suggest Pad diagrams, thereby promoting the standardization of IC footprint labeling.

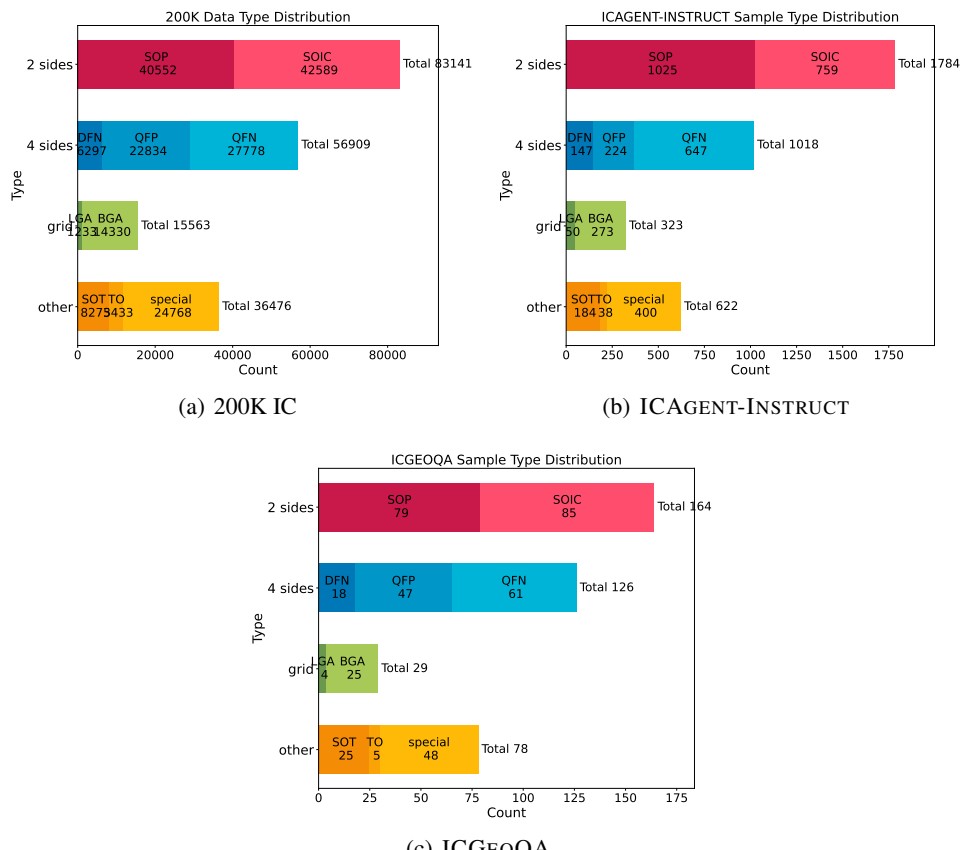

(a) 200K IC       (b) ICAGENT-INSTRUCT

(c) ICGEOQA

Figure A.1: IC package type distributions of datasets

## A.4 VQA TEMPLATES

The three agents in ICLABAGENT are trained to develop chain-of-thought (CoT) reasoning capabilities. To foster a robust and interpretable understanding of the three subtasks, the Visual Question Answering (VQA) templates are carefully designed. Expert-level PCB knowledge is incorporated into these VQA templates to train the agents as task-specific experts.

### A.4.1 DIAGRAM DETECTION DATASET

The Diagram Agent is trained to identify the target Suggest Pad diagram area within a datasheet page image. The target region is represented by a coordinate tuple $(x, y, width, height)$ normalized relative to the datasheet image size, where $x$ and $y$ denote the coordinates of the region's top-left corner, and $width$ and $height$ denote its dimensions. For datasheet pages containing a single Suggest Pad diagram, the prompt template is structured as follows:

---

**VQA Template Example for Diagram Detection (Single)**

| | |
|---|---|
| **Q** | <image>Suggest Pad is the recommended land pattern for an IC. There is only one Suggest Pad image in the picture. Please locate the Suggest Pad image and give the area of the Suggest Pad image in pure number pair (x,y,width,height) in proportion to the datasheet image size, where x and y are the coordinates of the left-upper corner of the area, width and height are the dimensions of the area. Note that (x,y)=(0,0) denotes the left upper corner of the image. |
| **A** | (0.35,0.14,0.29,0.44) |

---

A datasheet page may contain multiple Suggest Pad diagrams, all corresponding to a series of IC modules. When labeling a specific diagram, the target can be selected by specifying its position in the sequence of diagrams:

**VQA Template Example for Diagram Detection (Multiple)**

**Q** <image>Suggest Pad is the recommended land pattern for an IC. There are {count} Suggest Pad images in the picture. Please locate the {place} Suggest Pad image and give the area of the Suggest Pad image in pure number pair (x,y,width,height) in proportion to the datasheet image size, where x and y are the coordinates of the left-upper corner of the area, width and height are the dimensions of the area. Note that (x,y)=(0,0) denotes the left upper corner of the image.

**A** (0.35,0.14,0.29,0.44)

For simplicity, in the following sections, each datasheet page is assumed to contain only a single Suggest Pad diagram unless otherwise specified.

### A.4.2 PARAMETER PLANNING DATASET

To guide the Planning Agent's reasoning process, the parameter planning task is divided into two stages. In the first stage, the agent identifies the footprint arrangement type of the target IC diagram. In the second stage, it selects the relevant key parameters based on the type-specific checklist provided within the framework. The reasoning process of the Planning Agent is arranged into a **thinking** process enclosed within <think></think> and an **answering** process enclosed within <answer></answer>.

**VQA Template Example for Parameter Planning (2-sides)**

**Q** <image>Suggest Pad is the recommended land pattern for an IC. There is only one Suggest Pad image in the picture. The area of the Suggest Pad image is located by a pure number pair (x,y,width,height)=(0.35,0.14,0.29,0.44) in proportion to the datasheet image size, where x and y are the coordinates of the left-upper corner of the area, and width and height are the dimensions of the area. Note that (x,y)=(0,0) denotes the left upper corner of the image. Based on the located image, please classify this IC footprint as "2-sides", "4-sides", "grid" or "other".

**A** 2-sides

**Q** Based on the IC type, please choose the appropriate parameters to describe the positions and dimensions of the IC pins. Note that for "2-sides" ICs, parameters are chosen from: row, column, row spacing, column spacing, inner row spacing, outer row spacing, inner column spacing, outer column spacing, dx, dy, diameter; for "4-sides" ICs, parameters are chosen from: pin per side, side-to-side distance, inner side distance, outer side distance, pin spacing within side, dx1, dx2, dy1, dy2, dx, dy; for "grid" ICs, parameters are choosen from: row, column, row spacing, column spacing, dx, dy, diameter; for "other" ICs, parameters are: pin count, pin coordinates, pin dimensions. The reasoning process and answers are enclosed within <think> </think> and <answer> </answer> tags, respectively, i.e., <think> reasoning process here </think> <answer> answer here </answer>.

**A** <think>Let's think step by step. To identify the pins' positions, I will first count the rows and columns of the IC. Then, I will interpret the row spacings and column spacings of the IC. To identify the pins' dimensions, I will interpret the width and height of the pins. Finally, I will call "Tool1" and input the above parameters to generate the descriptions of this IC diagram. To sum up, I will find parameters: row, column, row spacing, column spacing, dx, dy.</think> <answer>IC type: 2-sides, extract parameters: row, column, row spacing, column spacing, dx, dy</answer>

**VQA Template Example for Parameter Planning (4-sides)**

**Q** <image>Suggest Pad is the recommended land pattern for an IC. There is only one Suggest Pad image in the picture. The area of the Suggest Pad image is located by a pure number pair (x,y,width,height)=(0.35,0.14,0.29,0.44) in proportion to the datasheet image size, where x and y are the coordinates of the left-upper corner of the area, and width and height are the dimensions of the area. Note that (x,y)=(0,0) denotes the left upper corner of the image. Based on the located image, please classify this IC footprint as "2-sides", "4-sides", "grid" or "other".

**A** 4-sides

**Q** Based on the IC type, please choose the appropriate parameters to describe the positions and dimensions of the IC pins. Note that for "2-sides" ICs, parameters are chosen from: row, column, row spacing, column spacing, inner row spacing, outer row spacing, inner column spacing, outer column spacing, dx, dy, diameter; for "4-sides" ICs, parameters are chosen from: pin per side, side-to-side distance, inner side distance, outer side distance, pin spacing within side, dx1, dx2, dy1, dy2, dx, dy; for "grid" ICs, parameters are choosen from: row, column, row spacing, column spacing, dx, dy, diameter; for "other" ICs, parameters are: pin count, pin coordinates, pin dimensions. The reasoning process and answers are enclosed within <think> </think> and <answer> </answer> tags, respectively, i.e., <think> reasoning process here </think> <answer> answer here </answer>.

**A** <think>Let's think step by step. To identify the pins' positions, I will first count the pins on each side. Then, I will interpret the center distance of two opposite sides and the spacing of pins of the same side. To identify the pins' dimensions, I will interpret the width and height of the pins. There is an additional pin in the center of the diagram, so there are two sets of pin dimensions. Finally, I will call "Tool2" and input the above parameters to generate the descriptions of this IC diagram. To sum up, I will find parameters: pin per side, side-to-side distance, pin spacing within side, dx1, dy1, dx2, dy2.</think> <answer>IC type: 4-sides, extract parameters: pin per side, side-to-side distance, pin spacing within side, dx1, dy1, dx2, dy2</answer>

**VQA Template Example for Parameter Planning (grid)**

**Q** <image>Suggest Pad is the recommended land pattern for an IC. There is only one Suggest Pad image in the picture. The area of the Suggest Pad image is located by a pure number pair (x,y,width,height)=(0.35,0.14,0.29,0.44) in proportion to the datasheet image size, where x and y are the coordinates of the left-upper corner of the area, and width and height are the dimensions of the area. Note that (x,y)=(0,0) denotes the left upper corner of the image. Based on the located image, please classify this IC footprint as "2-sides", "4-sides", "grid" or "other".

**A** grid

**Q** Based on the IC type, please choose the appropriate parameters to describe the positions and dimensions of the IC pins. Note that for "2-sides" ICs, parameters are chosen from: row, column, row spacing, column spacing, inner row spacing, outer row spacing, inner column spacing, outer column spacing, dx, dy, diameter; for "4-sides" ICs, parameters are chosen from: pin per side, side-to-side distance, inner side distance, outer side distance, pin spacing within side, dx1, dx2, dy1, dy2, dx, dy; for "grid" ICs, parameters are choosen from: row, column, row spacing, column spacing, dx, dy, diameter; for "other" ICs, parameters are: pin count, pin coordinates, pin dimensions. The reasoning process and answers are enclosed within <think> </think> and <answer> </answer> tags, respectively, i.e., <think> reasoning process here </think> <answer> answer here </answer>.

**A** <think>Let's think step by step. To identify the pins' positions, I will first count the rows and columns of the IC. Then, I will interpret the row spacings and column spacings of the IC. To identify the round pins' dimensions, I will interpret the diameter of the pins. Finally, I will call "Tool3" and input the above parameters to generate the descriptions of this IC diagram. To sum up, I will find parameters: row, column, row spacing, column spacing, and diameter.</think>
<answer>IC type: grid, extract parameters: row, column, row spacing, column spacing, diameter</answer>

**VQA Template Example for Parameter Planning (other)**

**Q**  <image>Suggest Pad is the recommended land pattern for an IC. There is only one Suggest Pad image in the picture. The area of the Suggest Pad image is located by a pure number pair (x,y,width,height)=(0.35,0.14,0.29,0.44) in proportion to the datasheet image size, where x and y are the coordinates of the left-upper corner of the area, and width and height are the dimensions of the area. Note that (x,y)=(0,0) denotes the left upper corner of the image. Based on the located image, please classify this IC footprint as "2-sides", "4-sides", "grid" or "other".

**A**  other

**Q**  Based on the IC type, please choose the appropriate parameters to describe the positions and dimensions of the IC pins. Note that for "2-sides" ICs, parameters are chosen from: row, column, row spacing, column spacing, inner row spacing, outer row spacing, inner column spacing, outer column spacing, dx, dy, diameter; for "4-sides" ICs, parameters are chosen from: pin per side, side-to-side distance, inner side distance, outer side distance, pin spacing within side, dx1, dx2, dy1, dy2, dx, dy; for "grid" ICs, parameters are choosen from: row, column, row spacing, column spacing, dx, dy, diameter; for "other" ICs, parameters are: pin count, pin coordinates, pin dimensions. The reasoning process and answers are enclosed within <think> </think> and <answer> </answer> tags, respectively, i.e., <think> reasoning process here </think> <answer> answer here </answer>.

**A**  <think>Let's think step by step. As this diagram is irregular, I will directly describe the pins in pin counts, pin coordinates, and pin diagrams. I will first count the number of pins. Then, I will interpret the coordinates of each pin. Finally, I will interpret the widths and heights of each pin. I will give the descriptions of this IC diagram in counts: pin count, pin coordinates: "name, x, y", and pin dimensions: "name, dx, dy".</think>
<answer>IC type: other, extract parameters:pin count, pin coordinates, pin dimensions</answer>

When standard key parameters are missing (*e.g.*, if "column spacing" is not explicitly labeled on the diagram for a "2-side" IC), the Planning Agent devises a calculation method to derive the missing parameter using the existing parameters displayed on the diagram. The following example illustrates a case in which the "column spacing" is absent from the diagram of a 2-side IC, while the "inner column spacing" and "outer column spacing" are provided instead.

**VQA Template Example for Parameter Planning (2-sides missing column spacing)**

**Q**  <image>Suggest Pad is the recommended land pattern for an IC. There is only one Suggest Pad image in the picture. The area of the Suggest Pad image is located by pure number pair (x,y,width,height)=(0.35,0.14,0.29,0.44) in proportion of the datasheet image size, where x and y are the coordination of the left-upper corner of the area, width and height are the dimension of the area. Note that (x,y)=(0,0) denotes the left upper corner of the image. Based on the located image, please classify this IC footprint as "2-sides", "4-sides", "grid" or "other".

**A**  2-sides

**Q**  Based on the IC type, please choose the appropriate parameters to describe the positions and dimensions of the IC pins. Note that for "2-sides" ICs, parameters are chosen from: row, column, row spacing, column spacing, inner row spacing, outer row spacing, inner column spacing, outer column spacing, dx, dy, diameter; for "4-sides" ICs, parameters are chosen from: pin per side, side-to-side distance, inner side distance, outer side distance, pin spacing within side, dx1, dx2, dy1, dy2, dx, dy; for "grid" ICs, parameters are choosen from: row, column, row spacing, column spacing, dx, dy, diameter; for "other" ICs, parameters are: pin count, pin coordinates, pin dimensions. The reasoning process and answers are enclosed within <think> </think> and <answer> </answer> tags, respectively, i.e., <think> reasoning process here </think> <answer> answer here </answer>.

*Continued on next page*

---

**VQA Template Example for Parameter Planning (2-sides missing column spacing) (Continued)**

---

**A**     <think>Let's think step by step. To identify the pins' positions, I will first count the rows and columns of the IC. Then, I will interpret the row spacings and column spacings of the IC. The column spacing is missing in this diagram, so I will find the inner and outer spacings of the two columns instead. To identify the pins' dimensions, I will interpret the width and height of the pins. Finally, I will call "Tool1" and input the above parameters to generate the descriptions of this IC diagram. To sum up, I will find parameters: row, column, row spacing, inner column spacing, outer column spacing, dx, dy.</think>
<answer>IC type: 2-sides, extract parameters: row, column, row spacing, inner column spacing, outer column spacing, dx, dy</answer>

---

### A.4.3    PARAMETER EXTRACTION DATASET

The Parameter Agent receives the Planning Agent's output as input instructions. These instructions generally comprise three steps: (1) extract the explicit parameter values specified by the Planning Agent, (2) compute any missing standard key parameters based on the planning instructions, and (3) compile standardized parameter values for use by downstream tools.

For example, in a "2-sides" IC, when all standardized parameters are explicitly provided on the diagram, the instruction VQA is simple:

---

**VQA Template Example for Parameter Extraction (All Key Parameters Available)**

**Q**    <image>The area of the Suggest Pad image is located by pure number pair (x,y,width,height)=(0.53,0.12,0.16,0.24) in proportion to the datasheet image size. Note that (x,y)=(0,0) denotes the left upper corner of the image. Based on the located suggest pad image, this IC can be classified as "2-sides". To describe the pins' positions and dimensions of this IC, please give the following pin parameters in pure numbers: row, column, row spacing, column spacing, dx, dy. The reasoning process and answers are enclosed within <think> </think> and <answer> </answer> tags, respectively, i.e., <think> reasoning process here </think> <answer> answer here </answer>.

**A**    <think>Find parameters: row=2, column=7, row spacing=6.1, column spacing=0.65, dx=0.45, dy=1.65.</think>
<answer>row=2, column=7, row spacing=6.1, column spacing=0.65, dx=0.45, dy=1.65</answer>

---

In another example, where the "row spacing" is not explicitly labeled in the diagram and only the "inner row spacing" is labeled, the (center) row spacing can be calculated based on the relationship between row spacing, inner row spacing, and the height of the IC pin $dy$:

$$\text{row spacing} = \text{inner row spacing} + dy \tag{1}$$

The VQA example in this case is shown as follows:

| VQA Template Example for Parameter Extraction (Rowing Spacing Omitted) | |
|---|---|
| Q | <image>The area of the Suggest Pad image is located by pure number pair (x,y,width,height)=(0.51,0.19,0.23,0.33) in proportion to the datasheet image size. Note that (x,y)=(0,0) denotes the left upper corner of the image. Based on the located suggest pad image, this IC can be classified as "2-sides". To describe the pins' positions and dimensions of this IC, please first find the following pin parameters in pure numbers: row, column, inner row spacing, column spacing, dx, dy. Then, calculate row spacing based on the equation "row spacing = inner row spacing + dy". Finally, give the pin parameters in pure numbers: row, column, row spacing, column spacing, dx, dy. The reasoning process and answers are enclosed within <think> </think> and <answer> </answer> tags, respectively, i.e., <think> reasoning process here </think> <answer> answer here </answer>. |
| A | <think>Find parameters: row=2, column=4, inner row spacing=7.49, column spacing=2.54, dx=1.78, dy=1.52. |
| | The row spacing is 7.49 + 1.52 = 9.01. |
| | Final parameters: row=2, column=4, row spacing=9.01, column spacing=2.54, dx=1.78, dy=1.52.</think> |
| | <answer>row=2, column=4, row spacing=9.01, column spacing=2.54, dx=1.78, dy=1.52</answer> |

All possible parameter reasoning cases are listed in Table A.1:

Table A.1: Possible parameter reasoning cases for each footprint arrangement type.

| type | standard parameters | missing parameters | reasoning |
|---|---|---|---|
| 2-sides | row, column row spacing column spacing dx, dy | row spacing | row spacing = (inner row spacing + outer row spacing)/2 = outer row spacing - dy = inner row spacing + dy |
| | | column spacing | column spacing = (inner column spacing + outer column spacing)/2 = outer column spacing - dx = inner column spacing + dx |
| 4-sides | pin per side, side-to-side distance, pin spacing within side, dx1, dy1, dx2, dy2 | side-to-side distance | side-to-side distance = (inner side distance + outer side distance)/2 = outer side distance - dx1 = inner side distance + dx1 |
| grid | row, column, row spacing, column spacing, diameter | missing pins | specify the missing pins by coordinates (*e.g.*, A1) |
| other | pin count, pin coordinates, pin dimensions | - | - |

## A.5 TRAINING DETAILS

The detailed training hyper-parameter settings of ICLABAGENT during the fine-tuning processes of the three task-specific agents are shown in Table A.2.

## A.6 LIMITATION

Due to limited training resources, this work employs only Supervised Fine-Tuning (SFT) to fine-tune the lightweight base model Qwen2-VL-7B using a small dataset. Despite this, it outperforms state-of-the-art general LMMs, such as GPT-5 and Gemini 2.5 Flash, by a substantial margin in

Table A.2: Hyper-parameter settings for fine-tuning three agents.

| Parameter | Value | | |
|---|---|---|---|
| | **Diagram Agent** | **Planning Agent** | **Parameter Agent** |
| Dataset | Diagram Detection | Parameter Planning | Parameter Extraction |
| Batch Size (per GPU) | 2 | 2 | 2 |
| Learning Rate (LR) | $5e^{-5}$ | $5e^{-5}$ | $5e^{-5}$ |
| LR Scheduler | cosine | cosine | cosine |
| Epoch | 3 | 3 | 3 |
| Cut-off Length | 4096 | 4096 | 4096 |
| Gradient Accumulation | 4 | 4 | 2 |
| Validation Set Ratio | 0.1 | 0.1 | 0.1 |
| Optimizer | AdamW | AdamW | AdamW |
| LoRA Rank | 8 | 8 | 8 |
| LoRA Alpha | 16 | 16 | 16 |
| LoRA Dropout | 0 | 0 | 0 |

the IC footprint labeling task. We recognize that Reinforcement Learning (RL) methods and feed-back refinements hold considerable potential for further improving ICLABAGENT's performance. Furthermore, larger and more advanced base models could enhance the framework's overall effectiveness. Nevertheless, our framework demonstrates the great potential of geometric reasoning in IC footprint labeling by introducing a human-like thinking process that decomposes the complex geometric reasoning task into simple, focused subtasks.

# B  DECLARATION OF LLM USAGE

We used large language models (LLMs) solely for light editing of prose—including wording refinement, grammar correction, and minor clarity improvements—in limited portions of this paper. All LLM-edited text was subsequently reviewed and revised by the authors.

