# OpenReview forum: "ICLabAgent: Multi-Agent LMM Framework for Integrated Circuit Footprint Geometry Labeling"
_ICLR.cc/2026/Conference — ICLR 2026 Conference Withdrawn Submission_

### Official Review · Reviewer_LGRX · 2025-10-25

**Soundness:** 2
**Presentation:** 3
**Contribution:** 3
**Rating:** 4
**Confidence:** 3

**Summary:**

The paper presents a significant and novel contribution to the application of LMMs in the critical engineering domain of Integrated Circuit footprint geometry labeling. The proposed ICLABAGENT framework successfully transitions from the existing end-to-end "black-box" approaches to an interpretable, decomposed multi-agent architecture that explicitly models the sequential cognitive workflow of expert PCB engineers. This structural stability results in a substantial improvement in overall geometric accuracy, establishing a new state-of-the-art performance level. The work is supported by robust component evaluation and an instructive ablation study demonstrating the non-redundant contribution of each specialized agent.

**Strengths:**

1.	The framework introduces an interpretable, decomposed multi-agent LMM architecture that explicitly models the four-stage sequential cognitive workflow of expert PCB engineers. This design fundamentally addresses the stability and black-box limitations of existing end-to-end models.
2.	The novel ICAGENT-INSTRUCT dataset provides rich, expert-level procedural supervision, including dynamic planning strategies, that enables the model to learn how to reason geometrically rather than relying only on final outputs .
3.	The framework achieves state-of-the-art overall layout accuracy, delivering a substantial improvement over the LLM4-IC8K baseline and meeting the strict precision requirements necessary for PCB manufacturing .
4.	Rigorous ablation studies confirm the non-redundant and essential contribution of each specialized agent in the workflow, validating the robustness and structural soundness of the decomposed design over end-to-end black-box approaches.

**Weaknesses:**

1.	The theoretical argument in Section 5.4 (Stage-3-only) for "constraining output space reduces cumulative errors" is insufficient and non-intuitive. This would elevate the finding from an specific observation to a general principle for structured reasoning tasks
2.	The paper omits critical "performance-cost balance" metrics for industrial utility, despite specifying GPU hardware (Section 5.1). It does not report total training time (e.g., hours on 8×A100) or inference time cost.
3.	It lacks a comparison with the LLM4-IC8K baseline in these metrics. Intuitively, multi-agent sequential reasoning may consume more computing resources , which raises questions about the fairness of direct performance comparison and limits assessment of its industrial deployment potential.
4.	The sole use of absolute error (MAE) fails to distinguish error characteristics across parameter type. It is recommended that the authors supplement the absolute error metrics with normalized errors.

**Questions:**

The technical attributes and working mechanism of the Description Generation Tool (Stage 4) are insufficiently described, leaving ambiguity as to whether it uses fixed algorithmic logic or model-based reasoning. If the tool uses fixed algorithmic logic (e.g., geometric rules), please specify the core logic for different footprint types.

---

### Official Review · Reviewer_j5Ne · 2025-10-31

**Soundness:** 3
**Presentation:** 3
**Contribution:** 2
**Rating:** 4
**Confidence:** 4

**Summary:**

This paper introduces ICLABAGENT that is a novel multi-agent framework leveraging large multimodal models for automated integrated circuit footprint geometry labeling. The framework emulates the step-by-step reasoning process of expert PCB engineers, addressing limitations of existing end2end black-box methods. It comprises three specialized agents: Diagram Agent, Planning Agent, and Parameter Agent. Then, a new dataset called ICAGENT-INSTRUCT, is constructed to train these agents with expert-level reasoning annotations. This work highlights the potential of multi-agent LMM systems in complex geometric reasoning tasks within EDA workflows.

**Strengths:**

- This work addresses a real-world bottleneck in PCB design: manual IC footprint labeling from datasheets is slow, error-prone, and lacks automation.
- Explicitly models human-expert reasoning, avoiding black-box shortcuts and improving trust and debuggability.
- This work gains significant accuracy gains and it outperforms GPT-5 by 94.6% and Gemini-2.5-Flash by > 50% on the same benchmark. Besides, it has +10.3% IoU over previous best and +79.5% over human industrial baseline.

**Weaknesses:**

- Limited generation validation: since all experiments conducted on a single benchmark (ICGEOQA), there is no evaluation on other public libraries (SnapEDA, Ultra Librarian) to assess domain robustness.
- Limited failure case discussion: here only high-level limitations mentioned, but no detailed qualitative examples of typical failure patterns.

**Questions:**

- When footprint diagrams and textual specs conflict, which source does the framework trust, and can it explicity model conflict detection and confidence weighting?
- Could injecting explicit symmetries, DRC rules, or overlap-area constraints import few-shot robustness compared with the current purely data-driven approach?
- How does the multi-agent inference cost compare with traditional CV-rule hybrid methods, and what model-compression/quantization steps maintain accuracy while reducing carbon footprint?
- Are this work plugins for Cadence/KiCad and what is the integration overhead?

---

### Official Review · Reviewer_toJk · 2025-11-01

**Soundness:** 1
**Presentation:** 2
**Contribution:** 1
**Rating:** 2
**Confidence:** 3

**Summary:**

This paper focuses on the problem of integrated circuit footprint geometry labeling, which is the process of extracting specific information about the integrated circuit footprint—such as the number of pins, coordinates of the pins, pin dimensions, etc.—from an integrated circuit datasheet. This paper introduces a new framework that leverages four different vision-language models as agents for different stages of the extraction process: (1) a diagram agent for locating the footprint diagram in the datasheet, (2) an agent for categorizing the footprint pattern and high-level characteristics of the design, (3) an agent for pulling out more fine-grained parameters of the design, and (4) an agent for specifying exactly the pin count, coordinates, and dimensions.

**Strengths:**

- Integrating vision-language models into EDA flows is a timely and promising area of research. This work also presents high novelty in addressing a stage of IC development that has not been touched on much in previous work.
- Splitting the IC footprint labeling task into multiple steps for different agents is a good idea to overcome the limitations of solving the task zero-shot.

**Weaknesses:**

- This could be due to my lack of background in the area, but it is very difficult to grasp the problem that this paper is trying to solve. The problem definition and motivation are not clearly established until late in the paper (see comments below).
- The experimental comparison to general-purpose LMMs is unfair. Testing these models only in a zero-shot setting provides a significant and potentially misleading advantage to the authors' fine-tuned model. A fair comparison would require at least few-shot prompting for the baselines.
- The paper needs to describe in greater detail the construction of its proposed dataset. From my understanding, the dataset is based on prior work, but the paper fails to clearly articulate the specific, novel contributions made during its construction. While the authors mention creating a 'dynamic planning and reasoning' dataset, the process for generating these new reasoning-focused annotations is not sufficiently detailed. This makes it difficult to assess the novelty and significance of this contribution.

**Questions:**

- Especially since this is a machine learning conference, it could be helpful to show a figure of what is meant by a footprint diagram and “Suggest Pads”/“Land Patterns.”
  The reader can look this up on the internet, but it would be helpful to have a small illustration, even in the appendix.

- > These steps are essential for ensuring interpretability and reliability. In contrast, our multi-agent design explicitly mirrors this human workflow through diagram detection, type classification, parameter extraction, and description generation.

  No workflow was mentioned at this point in the paper.

- Line 107: Since Figure 1 is being referenced, it would be good to have a brief overview in text so that readers can follow along with the figure.

- Lines 137–138: What is meant by perceiving–planning–reasoning–execution architecture here?

- Line 177: It would be really nice to have a figure at this point in the paper so we can better understand the problem that is being solved.

- Line 183: This explanation needs to appear much higher in the paper. Until this paragraph, the general problem that this paper aims to solve is extremely unclear.

- Line 208: Say where in Figure 2; otherwise it is difficult to cross-reference.

---

### Official Review · Reviewer_BkGe · 2025-11-02

**Soundness:** 3
**Presentation:** 3
**Contribution:** 2
**Rating:** 4
**Confidence:** 3

**Summary:**

This paper introduces ICLabAgent, a multi-agent large multimodal model (LMM) framework for automated IC footprint geometry labeling — a key task in PCB design that converts pin diagrams into machine-readable parameters. The framework mimics the human engineer’s reasoning process via three specialized agents: a Diagram Agent for region detection, a Planning Agent for footprint classification and parameter planning, and a Parameter Agent for parameter extraction and reasoning. The authors also introduce ICAgent-Instruct, a new dataset with expert-level planning annotations.

**Strengths:**

The paper presents a novel agentic decomposition of an engineering workflow that was previously handled as a black-box task. It is the first to apply multi-agent LMM reasoning to IC footprint labeling, bridging the gap between visual-semantic understanding and geometric reasoning. The modular design (region detection → type classification → parameter extraction) improves interpretability and reduces hallucination, validated via comprehensive ablation.  The ICAGENT-INSTRUCT dataset is carefully curated to capture step-by-step reasoning, enhancing reproducibility. The model consistently outperforms both general-purpose and domain-specific baselines on multiple granular metrics (IoU, MAE, RMSE). The work addresses a real industrial pain point, and the presented approach could generalize to other engineering diagram-understanding tasks.

**Weaknesses:**

All experiments are conducted on datasets derived from ICGEO8K. It is unclear whether the method generalizes to unseen schematic types (e.g., analog vs digital ICs, irregular layouts).

The ablation in Table 3 focuses on stage removal but lacks insight into cross-agent communication efficiency and error propagation.

The reported superiority over GPT-5 and Gemini-2.5 is informative but somewhat unfair since those models are used zero-shot; fine-tuned LMM baselines or strong visual backbones (e.g., InternVL) would strengthen the claim.

The paper references diagram-detection “tools” (e.g., Tool 1–3) without sufficient architectural or hyperparameter detail.

Limited discussion of computational cost and latency, which are critical in industrial PCB workflows.

**Questions:**

How is inter-agent communication implemented? Are messages exchanged in natural language or structured schema, and how is context preserved across agents?

What is the impact of CoT-style supervision versus simple instruction fine-tuning in each agent’s performance?

How does the model handle ambiguous or partially occluded footprint diagrams where geometric parameters are missing or inconsistent?

Could the authors clarify the robustness of the approach to noisy or low-resolution datasheets (common in older IC documents)?

Are there plans to release ICAGENT-INSTRUCT with standardized tool APIs to enable reproducible evaluation?

---

### Note · Authors · 2025-11-13

I have read and agree with the venue's withdrawal policy on behalf of myself and my co-authors.